# Correlation of Internal Exposure Levels of Polycyclic Aromatic Hydrocarbons to Methylation of Imprinting Genes of Sperm DNA

**DOI:** 10.3390/ijerph16142606

**Published:** 2019-07-22

**Authors:** Yufeng Ma, Zhaoxu Lu, Li Wang, Mei Qiang

**Affiliations:** Department of Children and Adolescences Health, School of Public Health, Shanxi Medical University, Taiyuan 030001, China

**Keywords:** polycyclic aromatic hydrocarbons, embryonic development, imprinted genes, methylation, sperm

## Abstract

Human exposure to polycyclic aromatic hydrocarbons (PAHs) results in adverse health implications. However, the specific impact of paternal preconception PAHs exposure has not been fully studied. In this study, a total of 219 men aged 24–53 were recruited and an investigation was conducted using a questionnaire requesting information about age, occupation, education, family history, lifestyle, and dietary preferences. Urine and semen samples were examined for the levels of the hydroxyl metabolites of PAHs (OH-PAHs) using ultra-high-performance liquid chromatography–tandem mass spectrometry and sperm DNA methylation by pyrosequencing. The results from the correlation analysis using seven OH-PAHs and the average methylation levels of the imprinting genes *H19*, *PEG3*, and *MEG3* indicated that 1-OHPH is positively correlated with *H19*/*PEG3* methylation levels. We further examined the correlation between each OH-PAH and the methylation levels at the individual CpGs. The results showed 1-OHPH is specifically correlated with CpG4 and CpG6 of the imprinted gene *H19*, CpG1 and CpG2 of *PEG3,* and CpG2 of *MEG3*; whereas 1-OHP is positively correlated with *PEG3* at CpG1. Multivariate regression model analysis confirmed that 1-OHPH and 1-OHP are independent risk factors for the methylation of *H19*. These data show that sperm DNA imprinting genes are sensitive to adverse environmental perturbations.

## 1. Introduction

The impact of environmental toxic agents on human reproduction and development has become increasingly concerning [1,2]. Polycyclic aromatic hydrocarbons (PAHs), a class of organic compounds, are known for their genetic toxicity and carcinogenic effects. PAHs are globally generated and distributed pollutants, which are commonly derived from the incomplete combustion and/or pyrolysis of fossil fuels, industrial or domestic coal, wood, cigarettes, and food items with which humans encounter through daily contact. As PAHs can cross the placental barrier, pregnant women and the developing fetus are particularly sensitive to environmental PAHs exposure. PAHs have been reported to have detrimental effects, including mutagenic, genotoxic, and reproductive effects, and causing developmental disorders [3,4,5]. Previous studies have shown that maternal exposure to PAHs during pregnancy increases the incidence of intrauterine growth retardation and preterm delivery [6]. Comparing the data from children′s brain tumors with the controls, parental occupational exposure to PAHs in the five years prior to birth increased the risk of brain tumors occurring in the child [7]. The levels of PAHs metabolites in serum were measured by collecting the peripheral blood of pregnant women with fetal malformations, and the umbilical cord blood of the fetuses or newborns. The levels of PAHs in umbilical cord blood were found to be higher in patients with neural tube defects. Therefore, PAHs may be associated with the occurrence of fetal birth defects. These findings suggest that PAHs are closely related to reproductive health and the incidence of adverse birth outcomes in the human population. Most of the previous studies reported the adverse effects of PAHs on fetus through direct maternal exposure. However, studies regarding paternal exposure to PAHs on birth defects are limited. In recent years, parental impacts have been gradually recognized. Studies have shown that environmental PAHs damage sperm DNA, which results in sperm chromatin instability [8]. DNA methylation is an important epigenetic mechanism that can be used to reveal the impact of previous paternal experience on the modification of imprinted genes [9]. Studies suggest that changes in sperm DNA methylations are associated with embryonic growth and development. Benchaib et al. analyzed the methylation status of the sperm DNA in 63 semen samples and found that DNA hypomethylation is closely related to pregnancy outcome [10]. Genomic imprinting refers to the epigenetic mechanisms that result in the monoallelic expression of a subset of genes in a parent-of-origin manner. The methylation of imprinted genes is a unique epigenetic modality that plays a specific role in fertility, placental function, and fetal growth and development [11]. The major genes involved in embryonic development include *IGF2*, *H19*, *Meg3*, *Peg1*/*Mest*, and *Peg3*. Epigenetic disruptions alter the specific dosage of imprinted genes, which can lead to various developmental abnormalities often affecting the fetal growth and health of the offspring. For example, Prader–Willi syndrome was the first human disease identified as genomic imprinting gene abnormalities [12]. These findings suggest that imprinted gene methylation may be an important epigenetic marker of detrimental factors affecting fetal development. However, the mechanism through which PAHs exposure affects sperm imprinting genes has not been reported.

Benzo [a] pyrene (B[a]P) is an important type of PAHs. However, its concentration is low compared to other PAHs even in B[a]P exposure in some occupational settings, limiting its use as a biomarker in tobacco smoke studies [13,14]. Excretory 1-hydroxypyene, a metabolite of pyrene, has been explored as a biological monitoring indicator of the body burden of PAHs [15,16,17]. PAHs are extensively metabolized by cytochrome P450 enzymes in humans and animals. The major metabolites of PAHs are monohydroxy phenols (hydroxylated metabolites of polycyclic aromatic hydrocarbons (OH-PAHs)) and dihydrodiols [18,19,20]. Biomonitoring of PAH metabolites is an important approach for evaluating human exposure and the body burden of PAHs. Therefore, these biomarkers have been widely used to characterize PAH exposure, with urinary OH-PAHs receiving considerable attention [21]. Therefore, we tested seven OH-PAHs in the urine to explore whether PAHs exposure would alter the methylation level of sperm imprinting genes, which may provide a theoretical basis for the study of the etiology and pathogenesis of birth defects.

## 2. Materials and Methods

### 2.1. Study Participants and Data Collection

In this study, a total of 219 men were enrolled from two clinics within reproductive health centers in Shanxi Province, China, during March–April 2015 and March 2016. The participants were chosen from those who visited for the purpose of either a pre-pregnancy check or family infertility examination. Upon enrollment, they were asked to complete a questionnaire including social-demographic factors: age, occupation, education, family history of birth defects or diseases, lifestyle, and dietary preferences. Those were excluded from the study if they had a family history of birth defects or heredopathia.

The study was conducted in accordance with the Declaration of Helsinki, and the protocols were approved by the Science and Research Ethics Committee of Shanxi Medical University Institutional Review Board (Project identification code: 2016LL027, date of approval: 10 March 2016). Written informed consent was obtained from each participant prior to enrolment in the study. 

### 2.2. Semen and Urine Collection

From each participant, we collected 50 mL midstream urine with clean containers. Semen samples were obtained by requiring participants to abstain from ejaculation for at least 3 days but no more than 7 days prior to their visit. Semen was collected on site by masturbation without the use of lubricants into a sterile polypropylene collection container. After liquefaction, semen samples were centrifuged at 1200 rpm for 15 min to pellet sperm and stored at −80 °C until subsequent DNA methylation analyses. All samples were stored at −80 °C prior to use.

### 2.3. Measurement of the Concentrations of Monohydroxylated PAHs (OH-PAHs) in Urine

The hydroxyl metabolites concentrations were determined using an assay with enzyme digestion and solid-phase extraction followed by ultra-high-performance liquid chromatography–tandem mass spectrometry (UPLC-MS/MS, Shimadzu, Kyoto, Japan) to quantify 7 OH-PAHs (1-and 2-OHN, 1-and 2-hydroxynaphthalene, 2-and 3-OHF, 2-and 3-hydroxyfluorene, 1-and 2-OHPH, 1-and 2-hydroxyphenanthrene, 1-OHP, and 1-hydroxypyrene) using a 10 mL urine sample. This method was previously described in detail [22,23]. 

For the enzymatic hydrolysis, frozen urine samples were melted at room temperature, centrifuged at 2000 rpm for 10 min, then 10.0 mL of each urine sample was accurately transferred into fresh 50-mL tubes. Then, 10.0 mL of acetic acid–ammonium acetate buffer solution (pH = 5.0) was incubated with 30 μL of β-glucuronidase (Sigma, New York, NY, USA), mixed well, added into the urine sample, and incubated overnight at 37 °C. Then, C18 SPE cartridges were activated with 5 mL of methanol and 5.0 mL of distilled water. The hydrolysate was enriched using a C18 solid-phase extraction column eluted with 5.0 mL of 30% methanol, then slowly eluted with 5.0 mL of methanol. The eluent was slowly dried using blown nitrogen and then fixed up to a volume of 600 μL with methanol. The concentrated solution was passed through a 0.45μm syringe filter into an autosampler amber vial for ultra-high-performance liquid chromatography–tandem mass spectrometry (LC-MS/MS) analysis. 

For the LC-MS/MS analysis of urine samples, we separated the urine sample constituents with ultra-high-performance liquid chromatography and injected subsequently into Waters Symmetry C18 column (250 mm × 4.6 mm, 5 μm). The column temperature was 35 °C and the flow rate was 500 μL/min. The program for gradient elution was set at 65%–100% methanol for 25 min, 100%–65% methanol for 3 min, and 65% methanol for another 2 min. The spectrometry measurements of the 7 OH-PAHs are shown in Table 1. 

### 2.4. DNA Extraction

Genomic DNA extraction was performed for each sperm sample by combining a modified sperm lysis protocol for a guanidinium thiocyanate method [24] and a QIAamp DNA micro kit (Qiagen, Valencia, CA, USA) for isolating DNA. The DNA was quantified using a Nanodrop 2000 Spectrophotometer. Data from participants who had sperm DNA-methylation data available (*n* = 174) were included in the correlation analysis. 

### 2.5. Bisulfite Treatment

Genomic DNA (1 µg) was treated with sodium bisulfite to convert unmethylated cytosine residues to uracil and leave methylated cytosine unchanged using an EZ Methylation Gold-Kit (Zymo Research, Orange, CA, USA) per the manufacturer’s instructions. The converted DNA was eluted in 15 μL of TE buffer (10 mM Tris-HCl, 0.1 mM EDTA, pH 7.5). Two microliters of the post-bisulfite-treated DNA were used for subsequent PCR amplification.

### 2.6. PCR Amplification of Bisulfite-Treated Sperm DNA and Subsequent Pyrosequencing

Bisulfite converted DNA (~50 ng) was amplified by PCR using a PyroMark PCR Kit (Qiagen, Valencia, CA, USA). All reactions were performed with the provided PCR mixtures (total volume 25 μL) with 0.2 μM each of the forward and reverse PCR primers. Each reaction also contained 2.5 μL of CoralLoad Concentrate (10×) for checking amplicons on an agarose gel. The reverse primer was conjugated to biotin. These single-stranded amplicons were isolated using Pyrosequencing Work Station and underwent pyrosequencing on a Pyromark Q96 MD pyrosequencing instrument (Qiagen, Valencia, CA, USA). The primers for PCR and pyrosequencing were described in a previous study [25]. The conditions for PCR were as follows: 94 °C for 15 min, followed by 45 cycles at 94 °C for 30 s, 56 °C for 30 s, 72 °C for 30 s, and a final 72 °C extension step for 10 min. 

The biotinylated PCR products were purified using streptavidin-sepharose beads (Amersham, Piscataway, NJ, USA) and sequenced using PyroMark Gold Q96 kit (Biotage AB, Uppsala, Sweden). The degree of methylation at each CpG site was determined using PyroMark CpG Software (Biotage AB, Uppsala, Sweden). Pyrosequencing assays were performed in duplicate in sequential runs (technical replicates), and the values shown represent the mean methylation for each individual CpG site. The number of CpGs sites analyzed at each differentially methylated region (DMR) was as follows: *H19* (7 CpGs), *MEG3* (8 CpGs), and *Peg3* (8 CpGs) [26]. 

### 2.7. Statistical Analysis

The results were analyzed with GraphPad Prism 5.0 (GraphPad Software, La Jolla, CA, USA). The continuous data are expressed as means ± standard errors (SE). Medians and interquartile ranges (IQR) were employed to describe distributions of the OH-PAHs concentrations. The Mann–Whitney U and Kruskal–Ruskal tests were used to compare the median levels of OH-PAHs between two or multiple social-demographic groups. Spearman’s rank correlation analysis was used to determine the correlation between methylation status of the 3 imprinting genes and urinary OH-PAHs concentrations. Finally, multivariate regression models were employed to analyze the relationship between levels of PAHs and changes in the methylation levels of the three imprinting genes *H19*, *MEG3*, and *PEG3* in sperm DNA when controlling for potential confounders. Correction for multiple comparisons was performed using Benjamini–Hochberg multiple testing to determine the false discovery rate (FDR). An FDR *q*-value < 0.05 was considered significantly different.

## 3. Results

### 3.1. Demographic Characteristics of the Study Participants

A questionnaire was administered to obtain general information about the 219 participants, as summarized in Table 2. The range of age was between 24 and 53 years old, with 50% of the participants being between 30 and 35 years of age. The educational level of 50% of the participants (*n* = 158) was high school (including secondary specialized schools) or above. There were 130 participants with three or more years of smoking experience. Only a few people (*n* = 39) like smoked food (more than once a week). Therefore, environmental PAHs exposure was likely through inhalation. No significant difference was found in the seven urinary OH-PAHs in the various demographic groups (*p* > 0.05). 

### 3.2. Distribution of Urine OH-PAHs among Different Demographic Groups

Urine samples were collected for urine PAHs metabolites measurement. In the current study, we examined seven OH-PAHs in urine samples. Table 3 shows the average levels normalized by urine gravity to correct for the influence derived from urinary concentration. The median level (25–75%) of the seven OH-PAHs was 0.0 for 2-OHN (0.0–0.428), 0.0 for 1-OHN (0.0–0.016), 0.004 for 3-OHF (0.0–0.039), 0.023 for 2-OHF (0.0–0.069), 0.129 for 2-OHPH (0.0–0.319), 0.015 for 1-OHPH (0.0–0.052), and 0.010 for 1-OHP (0.0–0.058). No significant difference was found for the seven urinary OH-PAHs among the different demographic groups (*p* > 0.05).

### 3.3. Levels of Sperm DNA Methylation of the Imprinting Genes

Three marks of imprinting genes, *H19*, *PEG3*, and *MEG3*, were selected and measured in this study. The average percentage of sperm DNA imprinting genes methylation from 174 of the 219 participants is shown in Figure 1. The average levels of methylation were as follows: *H19*, 86.74%; (95% CI: 86.05, 87.42); *PEG3*, 0.557% (95% CI: 0.286, 0.828); and *MEG3*, 2.267% (95% CI: 1.685, 2.850). 

### 3.4. Correlation Analysis between Concentrations of Urinary OH-PAHs and Average Levels of Sperm DNA Imprinting Genes Methylation

To examine the relationship between urinary OH-PAHs concentrations and methylation level of imprinted genes, correlation analyses were conducted using seven OH-PAHs and the average methylation level of each of the three genes. The results are shown in Table 4. 1-OHPH is positively correlated with the methylation levels of *H19* (coefficient, 0.189) and *PEG3* (coefficient, 0.190). However, this was not observed in the other six OH-PAHs. To investigate the correlation between OH-PAHs and specific CpG sites of each gene, we further examined the correlation between each OH-PAH and the methylation levels at the individual CpG level. The data in Table 5 show the correlation coefficient (*r*) of the CpG sites that were statistically significant (*p* < 0.05). 1-OHPH was positively correlated with CpG4 and CpG6 of the imprinted gene *H19* (*r* = 0.170 and 0.158, respectively); 1-OHPH was positively correlated with CpG1 and CpG2 of *PEG3* (*r* = 0.207 and 0.201, respectively) and positively correlated with CpG2 of the imprinting gene *MEG3* (*r* = 0.201); and 1-OHP and imprinting gene *PEG3* CpG1 were positively correlated (*r* = 0.156).

Finally, we performed a multiple stepwise linear regression analysis with the average methylation level of the imprinting genes. The potential confounders, such as age, education, drinking, smoking, and smoked food intake were selected as independent variables in the analysis with the seven OH-PAHs. The results summarized in Table 6 indicate that the average methylation levels of imprinting gene *H19* are associated with urinary levels of 1-OHP and 1-OHPH, which confirmed that 1-OHP and 1-OHPH levels are independent factors affecting the methylation level of *H19*.

## 4. Discussion

### 4.1. Sperm Chromatin is Sensitive to Environmental Pollutants

Increasing amounts of evidence are showing that environmental exposure to harmful agents including PAHs has adverse effects on reproduction and development in animal studies and human fetuses [26,27]. Birth defects are conditions existing at or before birth regardless of cause, including anatomical structure abnormalities, dysfunction, growth and development disorders, and metabolic abnormalities. Environmental factors are important contributors to birth defects. For example, among all causes of birth defects, 10% of anomalies were caused by a purely environmental factor. Among them, 25% of anomalies have a purely genetic cause, and environmental and genetic factors or unknown reasons accounted for the remaining 65% [28]. B[a]P is the main component of polycyclic aromatic hydrocarbons. In animal experiments, rats were treated with different concentrations of benzopyrene for four weeks. As a result, the length of the sperm telomere and benzopyrene concentration were found to be negatively related [29]. In 2014, 666 volunteers from Chongqing University (Chongqing, China) participated in a male reproductive health cohort study. The investigation found that urinary 1-OHPyr and 1-OHNap levels and sperm telomere length are negatively correlated, although no significant correlation was observed with semen quality or sperm apoptosis [29]. In the present study, we found a significant correlation between nonoccupational exposure urinary levels of 1-OHP, 1-OHPH and methylation of sperm DNA imprinting genes, suggesting that sperm chromatin is sensitive to PAHs.

### 4.2. Methylation of Imprinting Genes Is Affected by Adverse Environmental Perturbations 

Severe imprinting diseases, such as developmental disorders, are rare; however, the consequence of different severities of imprinting gene disorders on fetal growth and health outcomes is a concern to researchers. As a result, we investigated the relationship of imprinting methylation level with a common growth disorder, intrauterine growth restriction, and the potential role in regulating normal growth variation.

The *H19* gene is a growth regulatory gene in humans [12]. Widely expressed in the embryonic period, *H19* participates normal growth and development of embryos. *PEG3* is located on chromosome 19q13. The paternal expresses this gene, only from the father of the allele. Previous results showed that upregulation of Peg3 expression may be one of the causes of high birth weight [30]. Deregulation of the *DLK1-MEG3* imprinting cluster on chromosome 14q32 is thought to be responsible for the distinct phenotypes observed in the patients of maternal and paternal UPD14 syndromes, which is associated with pre- and postnatal growth restriction, premature puberty, and obesity [30]. Recent studies have suggested that environmental poisons are harmful to offspring′s health by altering epigenetic regulation. For instance, alcohol administration in pregnant mice affected the methylation patterns of five imprinting genes (*H19*, *Gtl2*, *Peg1*, *Snrpn*, and *PEG3*) in the somatic and sperm DNA of male offspring [14]. They observed a decrease in the number of methylated CpG in H19 spermatozoa, the number of methylated CpGs in H19 from the F2 progeny, and the mean sperm concentration [14].

A growing body of studies has shown the alteration of imprinting genes in response to various environmental differs. Among epigenetic markers, DNA methylation is the most studied mechanism as a reliable marker of development. It is dynamic and sensitive to internal and external environmental changes due to lifestyle factors, disease, and exposure to environmental toxicants [31]. Using the B[a]P chronic exposure male mice experimental model, our team found that B[a]P induced a significant reduction in the methylation levels of imprinting genes *H19* and *Meg3*, and significantly increased *Peg1*, *Peg3*, and small nuclear ribonucleoprotein-associated protein N (*SNRPN*) in sperm DNA [32]. The results in this study showed that the factors affecting the methylation level of the imprinted gene *H19* include urinary 1-OHP and 1-OHPH levels. These data support that imprinting genes are sensitive to environmental adverse perturbations and may be associated with a role of developmental plasticity. The imprinting plasticity response to environmental challenge may be tightly safeguarded in the face of environmental perturbations during fetal development. We, therefore, propose that imprinting genes may be potentially useful biomarkers for paternal environmental exposure experience, especially during preconception stages for in utero or early development, and for health outcomes later in life.

### 4.3. Association between Paternal PAH Exposure and Imprinting Genes

Direct environmental exposures have different impacts on exposed individuals in terms of somatic cell development. More importantly, epigenetic changes in the germ line (sperm or egg) are required to transmit epigenetic information to the next generation, which is termed epigenetic inheritance [33]. Therefore, paternal preconception environmental PAHs exposure can perturb the epigenetic modification of sperm. If the sperm is affected, this subsequently influences the phenotypic variation of the exposed individual through sperm DNA methylation, which can be used to address mechanisms underlying the effects of paternal environmental contaminants exposure on embryo development.

In the present report, we demonstrated a significant interaction effect of urinary levels of 1-OHP and 1-OHPH on methylation of sperm DNA imprinting genes. Although the mechanisms are far from being fully understood, this altered epigenetic modification may have adverse effects on fetal development. Consequently, whether 1-OHP and 1-OHPH are good urinary biomarkers of exposure to environment warrants further investigation.

## 5. Conclusions

These data indicate a significant association between the concentrations of urinary 1-OHP and 1-OHPH and the levels of methylation of sperm DNA imprinting genes, which provide evidence that sperm DNA imprinting genes are sensitive to adverse environmental perturbations. Since paternal preconception PAHs exposure may have adverse effects on fetal development, we, therefore, propose that sperm imprinting genes may be a potential biomarker for the screening of birth defects. Further follow-up studies of birth outcomes are needed as the extension of this study.

## Figures and Tables

**Figure 1 ijerph-16-02606-f001:**
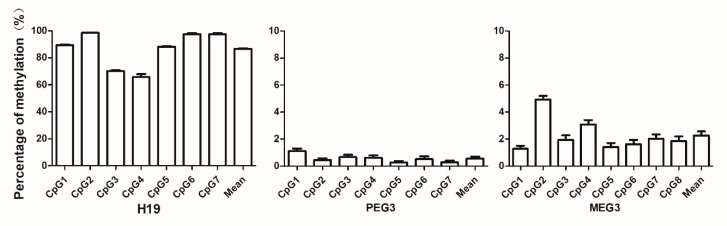
Levels of sperm DNA methylation at the differentially methylated regions (DMRs) of imprinting genes. The bars represent the percentage of methylated genes by individual CpG site for each DMR of the imprinting gene: *H19*, *Peg3*, and *Meg3*. The results are the means ± SEM of 219 human sperm samples.

**Table 1 ijerph-16-02606-t001:** The spectrometry measurement parameters of seven hydroxylated metabolites of polycyclic aromatic hydrocarbons (OH-PAHs).

Compound	Retention Time (min)	Parent Ion (*m/z*)	Daughter Ion (*m/z*)	Collision Energy (eV)
2-OHN	11.33	143	115	30
1-OHN	12.08	143	115	30
3-OHF	15.08	181	180	30
2-OHF	15.50	181	180	30
2-OHPH	16.49	193	165	30
1-OHPH	17.54	193	171	30
1-OHP	20.83	217	189	35

**Table 2 ijerph-16-02606-t002:** Demographic characteristics of the participants and urinary PAHs level (μg/mL).

Groups	*N*	2-OHN	1-OHN	3-OHF	2-OHF	2-OHPH	1-OHPH	1-OHP
		Median (25%–75%)	Median (25%–75%)	Median (25%–75%)	Median (25%–75%)	Median (25%–75%)	Median (25%–75%)	Median (25%–75%)
Age								
24–29	61	0.010 (0–0.701)	0 (0–0.215)	0.005 (0–0.044)	0.023 (0–0.076)	0.166 (0–0.351)	0.023 (0.003–0.070)	0.009 (0–0.071)
30–32	61	0 (0–0.253)	0 (0–0.009)	0.001 (0–0.043)	0.012 (0–0.047)	0.090 (0–0.250)	0.013 (0–0.040)	0.009 (0–0.039)
33–35	53	0 (0–0.178)	0 (0–0.125)	0.005 (0–0.053)	0.014 (0–0.076)	0.117 (0–0.340)	0.013 (0–0.050)	0.017 (0–0.069)
>35	44	0.016 (0–0.709)	0.005 (0–0.408)	0.006 (0–0.030)	0.033 (0–0.094)	0.145 (0–0.450)	0.026 (0.011–0.075)	0.014 (0–0.070)
Education								
Primary School or Below	4	0.196 (0.034–4.908)	0 (0–0.148)	0 (0–0.005)	0.024 (0–0.121)	0.023 (0–1.083)	0.036 (0.005–0.172)	0.006 (0.001–0.046)
Junior high	57	0.011 (0–0.658)	0 (0–0.024)	0.006 (0–0.049)	0.027 (0–0.087)	0.138 (0–0.332)	0.010 (0.001–0.047)	0.019 (0–0.075)
High school	56	0.003 (0–0.739)	0.006 (0–0.042)	0.035 (0–0.054)	0.028 (0–0.078)	0.095 (0–0.275)	0.015 (0–0.065)	0.010 (0–0.054)
College and above	102	0 (0–0.084)	0 (0–0.009)	0.035 (0–0.028)	0.015 (0–0.061)	0.146 (0–0.332)	0021 (0.003–0.048)	0.008 (0–0.052)
Smoking (years)								
0	80	0 (0–0.160)	0 (0–0.009)	0.005 (0–0.030)	0.014 (0–0.050)	0.134 (0.007–0.367)	0.018 (0.006–0.050)	0.009 (0–0.057)
1–3	9	0 (0–0.128)	0 (0–0.004)	0.009 (0.001–0.104)	0.025 (0.001–0.080)	0.211 (0.094–0.352)	0.010 (0.003–0.060)	0.008 (0–0.038)
3–5	17	0.010 (0–0.389)	0 (0–0.006)	0 (0–0.028)	0.011 (0–0.059)	0.091 (0–0.201)	0.010 (0–0.060)	0 (0–0.011)
6–10	43	0 (0–1.234)	0 (0–0.040)	0 (0–0.042)	0.014 (0–0.083)	0.129 (0–0.249)	0.014 (0–0.051)	0.011 (0–0.052)
>10	70	0.039 (0–0.504)	0.006 (0–0.036)	0.004 (0–0.040)	0.035 (0–0.100)	0.075 (0–0.317)	0.018 (0–0.061)	0.019 (0–0.074)
Drinking (years)								
0	134	0 (0–0.427)	0 (0–0.020)	0.002 (0–0.038)	0.025 (0–0.073)	0.162 (0–0.366)	0.028 (0.002–0.058)	0.013 (0–0.060)
0–3	7	0 (0–0.573)	0 (0–0.005)	0.018 (0–0.213)	0 (0–0.050)	0 (0–0.508)	0.014 (0–0.080)	0 (0–0.060)
3–5	9	0.067 (0.003–4.017)	0 (0–0.184)	0.007 (0–0.092)	0 (0–0.131)	0.036 (0–0.191)	0.025 (0–0.064)	0.001 (0–0.092)
6–10	28	0 (0–0.776)	0 (0–0.026)	0.006 (0–0.079)	0.003 (0–0.068)	0.071 (0–0.294)	0.009 (0–0.038)	0.011 (0–0.065)
>10	41	0 (0–0.095)	0 (0–0.016)	0.003 (0–0.026)	0.014 (0–0.049)	0.100 (0–0.243)	0.011 (0.003–0.032)	0.009 (0–0.042)
Bacon *								
0	180	0 (0–0.536)	0 (0–0.020)	0.002 (0–0.031)	0.024 (0–0.062)	0.141 (0–0.329)	0.016 (0–0.054)	0.009 (0–0.056)
1–2	32	0 (0–0.191)	0 (0–0.007)	0.007 (0–0.089)	0.018 (0–0.079)	0.078 (0–0.233)	0.011 (0–0.037)	0.018 (0–0.074)
3–4	6	0 (0–0.1152)	0 (0–0.014)	0.024 (0–0.295)	0.005 (0–0.066)	0 (0–0.748)	0.027 (0.003–0.065)	0.013 (0.003–0.063)
≧5	1	-	-	-	-	-	-	-

* Times/week.

**Table 3 ijerph-16-02606-t003:** The average concentrations of seven OH-PAHs (μg/L).

OH-PAH	*N*	Min.	Max.	25%	Median	75%
2-OHN	219	0.000	98.346	0.000	0.000	0.428
1-OHN	219	0.000	1.996	0.000	0.000	0.016
3-OHF	219	0.000	0.944	0.000	0.004	0.039
2-OHF	219	0.000	0.881	0.000	0.023	0.069
2-OHPH	219	0.000	4.224	0.000	0.129	0.319
1-OHPH	219	0.000	0.418	0.000	0.015	0.052
1-OHP	219	0.000	1.837	0.000	0.010	0.058

**Table 4 ijerph-16-02606-t004:** The correlation coefficients of OH-PAHs and the average methylation levels of imprinting genes.

OH-PAH	*H19*	*PEG3*	*MEG3*
2-OHN	−0.027	−0.041	−0.028
1-OHN	0.029	0.012	0.021
3-OHF	0.097	0.012	0.061
2-OHF	0.125	−0.006	−0.067
2-OHPH	0.102	0.034	0.026
1-OHPH	0.189 *	0.190 *	0.130
1-OHP	−0.127	0.147	0.054

Note: * denotes *p* < 0.05.

**Table 5 ijerph-16-02606-t005:** The correlation of urinary OH-PAHs and methylation level at individual CpG sites of imprinting genes.

Gene Sites	1-OHPH		1-OHP	
	*r*	*p*	*r*	*p*
H19-CpG4	0.170	0.024	-	-
H19-CpG6	0.158	0.037	-	-
PEG3-CpG1	0.207	0.006	0.156	0.034
MEG3-CpG2	0.201	0.008	-	-

**Table 6 ijerph-16-02606-t006:** Multiple linear regression analysis of *H19*.

	B	SE	t	q (FDR)
1-OHP	−3.62	1.05	−3.43	<0.01
1-OHPH	17.72	6.63	2.67	0.01

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
