# Peer review of "Correlation of Internal Exposure Levels of Polycyclic Aromatic Hydrocarbons to Methylation of Imprinting Genes of Sperm DNA"

_ijerph, 2019, doi:10.3390/ijerph16142606_

Round 1
Reviewer 1 Report
As a result, the author suggested a new future biomarker in gene level. It is very nice idea. But we all know that the gene level is hard to support evidences. In this paper, the evidence looks good and enough for their result. But I still worry about that it is a real story or not.
So I think, the joural can publish this to readers and they will give the final decison.
before publish, the conclusion is better for rewrite, it is to simple for your work summary. And also, check the grammar mistakes clearly, such as references style.
Author Response
1. Before publish, the conclusion is better for rewrite, it is too simple for your work summary.
R.: We accept the reviewer’s criticism and suggestion. We therefore have modified the conclusion. These changes can be seen in the revised manuscript. We hope that the reviewer would agree.
2. Check the grammar mistakes clearly.
R.: We thank the review’s suggestion and have had this manuscript polished by English edition. (Please see the certification attached)
Reviewer 2 Report
1. The article is poorly written as is illustrated by the extensive list of edits below. The latter is incomplete. Clearly, an extensive rewrite is required.
The following lines/wording need attention and indicate that the manuscript requires strong editing: L10, replace ‘on’ with to; L13, 'using questionnaire including age' is awkward; L17, a correlation analysis; L18, H19 and PEG3 ; L30-31, poor wording; L32, what does multiphasic refer to here?; L37, the phrase "as the results" makes no sense; L48/49, 'to the maternal direct exposure of fetal' incorrect & awkward; L39-40 suggest: Previous studies have shown that maternal exposure to PAHs during pregnancy resulted in an increase in the incidence of …; L40-42 also need attention; L53, "experience of father’s life on the imprinted genes modification" is awkward; L63-64,"a causal factor by genomic imprinting gene abnormalities" & difficult to understand; L67, poor expression: "B[a]P is an important pollutant of PAHs"; L75, "Therefore, we test" is awkward and has the wrong tense; L82, "visited for the purpose either pre-pregnant check" is poorly expressed; L84-5 "the information such as lifestyles..” poor wording; L92, need the plural form of was; L93, 'abtains' should be "to abstain"; L103 "simultaneously using" does not make sense; L111, "The nitrogen blown…" means what?; L120, "spectrometry parameters mass of 7 OH-PAHs in Table 1" and in the title of Table 1 do not make sense; L119-120, the wording "The spectrometry parameters mass of 7 OH-PAHs in Table 1" needs a verb; L120-121, the last sentence of this paragraph seems incomplete; L128, opening a sentence with a number is not a good choice and the sentence could be improved, thus: Data from participants who had sperm DNA-methylation data available (n = 174) were included in the correlation analysis.; L151, DMR is not defined; L161, should be 'changes in'; L167, suggest: A questionnaire was employed to obtain general information about the 219 participants.; L172-3, poor wording: ‘There is no significant difference was found’; L178, should be 0.0; L180, poor wording: There is no significant difference was found in 7…; L189 &192, concentrations would be better than levels; L202, suggest: The data in Table 5 show; L224, poor wording: Recent year, increase evidence that elevate..; L228/229, poor wording: It is proved that environmental factors are the important of birth defects, among other causes of birth defects, 10% of anomalies have…; L233, need space between weeks & As; L234, poor/incorrect wording: ‘concentration were dose-negative Related [29]’; L247, need plural form and space, thus: ‘humans and [12]’; L249 suggest to replace ‘of the imprinting gene, only from the father of the allele.’ with ‘of this gene’; L250, should be ‘up-regulation of Peg3 expression …’; L260, should be ‘studies’; L264, need ‘the’ before ‘BaP’; L265, ‘significantly reduce’ should be ’significant reduction in’; L268, should be ‘genes’; L275, suggest replacing ‘result in various influence on exposed individual.’ with ‘impact exposed individuals in various ways’; L278, need a space before [33]; L284, replace the comma with ‘and’ and remove ‘respectively,’; L286/7, suggest: ‘Consequently whether 1-OHP and 1-OHPH are good urinary biomarkers of exposure warrant further investigation.’; L290, suggest ‘.. imprinting genes and thereby provide evidence that the latter are sensitive to adverse environmental perturbations.’; L293-9, the required information is missing.
2. There is unnecessary repetition of the tabulated results in the text: see L178-181and L201-207.
Author Response
1. The article is poorly written as is illustrated by the extensive list of edits below. which indicates that the manuscript requires strong editing:
L10, replace ‘on’ with to; L13, 'using questionnaire including age' is awkward;L17, a correlation analysis; L18, H19 and PEG3 ; L30-31, poor wording; L32, what does multiphasic refer to here?; L37, the phrase "as the results" makes no sense; L48/49, 'to the maternal direct exposure of fetal' incorrect & awkward; L39-40 suggest: Previous studies have shown that maternal exposure to PAHs during pregnancy resulted in an increase in the incidence of …; L40-42 also need attention; L53, "experience of father’s life on the imprinted genes modification" is awkward; L63-64,"a causal factor by genomic imprinting gene abnormalities" & difficult to understand; L67, poor expression: "B[a]P is an important pollutant of PAHs"; L75, "Therefore, we test" is awkward and has the wrong tense; L82, "visited for the purpose either pre-pregnant check" is poorly expressed; L84-5 "the information such as lifestyles..” poor wording; L92, need the plural form of was; L93, 'abtains' should be "to abstain"; L103 "simultaneously using" does not make sense; L111, "The nitrogen blown…" means what?; L120, "spectrometry parameters mass of 7 OH-PAHs in Table 1" and in the title of Table 1 do not make sense; L119-120, the wording "The spectrometry parameters mass of 7 OH-PAHs in Table 1" needs a verb; L120-121, the last sentence of this paragraph seems incomplete; L128, opening a sentence with a number is not a good choice and the sentence could be improved, thus: Data from participants who had sperm DNA-methylation data available (n = 174) were included in the correlation analysis.; L151, DMR is not defined; L161, should be 'changes in'; L167, suggest: A questionnaire was employed to obtain general information about the 219 participants.; L172-3, poor wording: ‘There is no significant difference was found’; L178, should be 0.0; L180, poor wording: There is no significant difference was found in 7…; L189 &192, concentrations would be better than levels; L202, suggest: The data in Table 5 show; L224, poor wording: Recent year, increase evidence that elevate..; L228/229, poor wording: It is proved that environmental factors are the important of birth defects, among other causes of birth defects, 10% of anomalies have…; L233, need space between weeks & As; L234, poor/incorrect wording: ‘concentration were dose-negative Related [29]’; L247, need plural form and space, thus: ‘humans and [12]’; L249 suggest to replace ‘of the imprinting gene, only from the father of the allele.’ with ‘of this gene’; L250, should be ‘up-regulation of Peg3 expression …’; L260, should be ‘studies’; L264, need ‘the’ before ‘BaP’; L265, ‘significantly reduce’ should be ’significant reduction in’; L268, should be ‘genes’; L275, suggest replacing ‘result in various influence on exposed individual.’ with ‘impact exposed individuals in various ways’; L278, need a space before [33]; L284, replace the comma with ‘and’ and remove ‘respectively,’; L286/7, suggest: ‘Consequently whether 1-OHP and 1-OHPH are good urinary biomarkers of exposure warrant further investigation.’; L290, suggest ‘. imprinting genes and thereby provide evidence that the latter are sensitive to adverse environmental perturbations.’; L293-9, the required information is missing.
R.: We thank this reviewer for his/her nicely corrections. We have changed/corrected accordingly in each point, where they appear in the text. Since they are many items, we don’t describe one by one, please see they in the attached revised text before English editing.
Reviewer 3 Report
The manuscript needs extensive editing in terms of language. It is difficult to understand what the authors are trying to convey.
Author Response
The manuscript needs extensive editing in terms of language. It is difficult to understand what the authors are trying to convey.
R.: We thank the review’s suggestion and have had this manuscript polished by English edition. (Please see the certification attached)

Round 2
Reviewer 3 Report
Authors have addressed the concerns raised. the manuscript may be accepted after minor grammatical corrections and typos.